# Efficient Method for Bi-level Optimization with Non-smooth Lower-Level problem

## Abstract

Bi-level optimization plays a key role in a lot of machine learning applications. Existing state-of-the-art bi-level optimization methods are limited to smooth or some specific non-smooth lower-level problems. Therefore, achieving an efficient algorithm for the bi-level problems with a generalized non-smooth lower-level objective is still an open problem. To address this problem, in this paper, we propose a new bi-level optimization algorithm based on smoothing and penalty techniques. Using the theory of generalized directional derivative, we derive new conditions for the bilevel optimization problem with nonsmooth, perhaps non-Lipschitz lower-level problem, and prove our method can converge to the points satisfying these conditions. We also compare our method with existing state-of-the-art bi-level optimization methods and demonstrate that our method is superior to the others in terms of accuracy and efficiency.

## 1 Introduction

Bi-level optimization (BO) (Bard, 2013; Colson et al., 2007) plays a central role in various machine learning applications including hyper-parameter optimization (Pedregosa, 2016; Bergstra et al., 2011; Bertsekas, 1976), meta-learning (Feurer et al., 2015; Franceschi et al., 2018; Rajeswaran et al., 2019), reinforcement learning (Hong et al., 2020; Konda & Tsitsiklis, 2000). It involves a competition between two parties or two objectives, and if one party makes its choice first it will affect the optimal choice for the other party. Several approaches, such as Bayesian optimization (Klein et al., 2017), random search (Bergstra & Bengio, 2012), evolution strategy (Sinha et al., 2017), gradient-based methods (Pedregosa, 2016; Maclaurin et al., 2015; Swersky et al., 2014), have bee proposed to solve BO problems, among which gradient-based methods have become the mainstream for large-scale BO problems.

The key idea of the gradient-based method is to approximate the gradient of upper-level variables, called hypergradient. For example, the implicit differentiation methods (Pedregosa, 2016; Rajeswaran et al., 2019) use the first derivative of the lower-level problem to be 0 to derive the hypergradient. The explicit differentiation methods calculate the gradient of the update rules of the lower-level based on chain rule (Maclaurin et al., 2015; Domke, 2012; Franceschi et al., 2017; Swersky et al., 2014) to approximate the hypergradient. Mehra & Hamm (2019) reformulate the bilevel problem as a single-level constrained problem by replacing the lower level problem with its first-order necessary conditions, and then solve the new problem by using the penalty method. Obviously, all these methods need the lower-level problem to be smooth.

However, in many real-world applications, such as image restoration (Chen et al.; Nikolova et al., 2008), variable selection (Fan & Li, 2001; Huang et al., 2008; Zhang et al., 2010) and signal processing (Bruckstein et al., 2009), the objective may have a complicated non-smooth, perhaps non-Lipschitz term (Bian & Chen, 2017). Traditional methods cannot be directly used to solve the bilevel problem with such a lower-level problem. To solve the BO problems with some specific nonsmooth lower-level problems, researchers have proposed several algorithms based on the above-mentioned methods. Specifically, Bertrand et al. (2020) searched the regularization parameters for LASSO-type problems by approximating the hypergradient from the soft thresholding function (Donoho, 1995; Bredies & Lorenz, 2008; Beck & Teboulle, 2009). Frecon et al. (2018) proposed a primal-dual FMD-based method, called FBBGLasso, to search the group structures of group-LASSO problems. Okuno et al. (2021) used the smoothing method and constrained optimization method to search the regularization

Table 1: Representative gradient-based bi-level optimization methods.

| Method | Reference | Problem | Method type |
|--------|-----------|---------|-------------|
| FMD | Franceschi et al. (2017) | Smooth | Bi-level |
| RMD | Franceschi et al. (2017) | Smooth | Bi-level |
| Approx | Pedregosa (2016) | Smooth | Bi-level |
| Penalty | Mehra & Hamm (2019) | Smooth | Single-level |
| FBBGL | Frecon et al. (2018) | Group LASSO | Bi-level |
| SparseHO | Bertrand et al. (2020) | LASSO-type | Bi-level |
| SMNBP | Okuno et al. (2021) | $p$-norm | Single-level |
| SPNBO | Ours | Generalized | Single-level |

parameter of $q$-norm ($0 < q \leq 1$) and provided the convergence analysis of their method. We summarize several representative methods in Table 1. Obviously, all these methods and their theoretic analysis only focus on some specific problem and can not be used to solve the bilevel problem with a generalized nonsmoothed lower-level problem. Therefore, how to solve the BO problem with a generalized non-smooth lower-level objective and obtain its convergence analysis are still open problems.

To address this problem, in this paper, we propose a new algorithm, called SPNBO, based on smoothing (Nesterov, 2005; Chen et al., 2013) and penalty (Wright & Nocedal, 1999) techniques. Specifically, we use the smoothing technique to approximate the original non-Lipschitz lower-level problem and generate a sequence of smoothed bi-level problems. Then, a single-level constrained problem is obtained by replacing the smoothed lower-level objective with its first-order necessary condition. For each given smoothing parameter, we propose a stochastic constraint optimization method to solve the single-level constrained problem to avoid calculating the Hessian matrix of the lower-level problem. Theoretically, using the theory of generalized directional derivative, we derive new conditions for the bilevel optimization problem with nonsmooth, perhaps non-Lipschitz lower-level problem, and prove our method can converge to the points satisfying these conditions. We also compare our method with several state-of-the-art bi-level optimization methods, and the experimental results demonstrate that our method is superior to the others in terms of accuracy and efficiency.

**Contributions.** We summarize the main contributions of this paper as follows:

1. We propose a new method to solve the non-Lipschitz bilevel optimization problem based on the penalty method and smoothing method. By using the stochastic constraint method, our method can avoid calculating the Hessian matrix of the lower-level problem, which makes our method a lower time complexity.

2. Based on the Clarke generalized directional derivative, we propose new conditions for the bilevel problem with a generalized non-smoothed lower-level problem. We prove that our method can converge to the proposed conditions.

## 2 PRELIMINARIES

### 2.1 FORMULATION OF NON-SMOOTH BI-LEVEL OPTIMIZATION PROBLEM

In this paper, we consider the following non-smooth bi-level optimization problem:

$$\min_{\boldsymbol{\lambda}} f(\boldsymbol{w}^*, \boldsymbol{\lambda}) \quad s.t. \quad \boldsymbol{w}^* \in \arg\min_{\boldsymbol{w}} g(\boldsymbol{w}, \bar{\boldsymbol{\lambda}}) + \exp(\lambda_1)\varphi(h(\boldsymbol{w})), \tag{1}$$

where $\boldsymbol{\lambda} := [\lambda_1, \lambda_2, \cdots, \lambda_m]^T \in \mathbb{R}^m$, $\bar{\boldsymbol{\lambda}} := [\lambda_2, \cdots, \lambda_m]^T$ and $\boldsymbol{w} \in \mathbb{R}^d$. $f : \mathbb{R}^d \times \mathbb{R}^m \mapsto \mathbb{R}$ and $g : \mathbb{R}^d \times \mathbb{R}^m \mapsto \mathbb{R}$ are twice continuously differentiable on $\boldsymbol{w}$ and $\boldsymbol{\lambda}$. $\varphi(\cdot) : \mathbb{R}^n \mapsto \mathbb{R}$ is twice continuously differentiable. $h(\cdot) : \mathbb{R}^d \mapsto \mathbb{R}^n$ is continuous, not necessarily convex, not differentiable, or even not Lipschitz at some points. Assume $h(\boldsymbol{w}) := (h_1(\boldsymbol{D}_1^T\boldsymbol{w}), h_2(\boldsymbol{D}_2^T\boldsymbol{w}), \cdots, h_n(\boldsymbol{D}_n^T\boldsymbol{w}))$, where $\boldsymbol{D}_i \in \mathbb{R}^{d \times r}$ and $h_i : \mathbb{R}^d \mapsto \mathbb{R}$ ($i = 1, 2, \cdots, n$) is continuous. For a fixed point $\bar{\boldsymbol{w}}$, assume we have an index set $\mathcal{I}_{\bar{\boldsymbol{w}}} = \{i \in \{1, 2, \cdots, n\} : h_i \text{ is not Lipschitz continuous at } \boldsymbol{D}_i^T\bar{\boldsymbol{w}}\}$ and if $i \notin \mathcal{I}_{\bar{\boldsymbol{w}}}$, $h_i$ is twice continuously differentiable.

## 2.2 EXAMPLES OF NON-SMOOTH NON-LIPSCHITZ LOWER-LEVEL PROBLEMS

The non-smooth non-Lipschitz optimization problems widely exist in image restoration (Chen et al.; Nikolova et al., 2008), variable selection (Fan & Li, 2001; Huang et al., 2008; Zhang et al., 2010) and signal processing (Bruckstein et al., 2009). Here, we give two examples as follows.

1. $l_p$-norm (Chen et al., 2013): $\min_{\boldsymbol{w}} g(\boldsymbol{w}, \bar{\boldsymbol{\lambda}}) + \exp(\lambda_1) \sum_{i=1}^{d} |\boldsymbol{w}_i|^p$, where $p \in (0, 1]$.

2. OSCAR penalty (Bondell & Reich, 2008): $\min_{\boldsymbol{w}} g(\boldsymbol{w}, \bar{\boldsymbol{\lambda}}) + \exp(\hat{\lambda})\|\boldsymbol{w}\|_1 + \exp(\check{\lambda}) \sum_{i<j} \max\{\boldsymbol{w}_{\mathcal{G}_i}, \boldsymbol{w}_{\mathcal{G}_j}\}$, where $\mathcal{G}_i$ denotes the group index.

Note that Okuno et al. (2021) only considered the bilevel problem with the lower-level problem given in Example 1. Their theoretical analysis is not suitable for the problem in Example 2 or even more complicated formulation.

# 3 PROPOSED METHOD

In this section, we give a brief review of the smoothing method and then propose our stochastic gradient algorithm based on the penalty method and single-level reduction method to solve the bilevel problem.

## 3.1 SMOOTHING TECHNIQUE

Here, we give the definition of smoothing function (Nesterov, 2005; Chen et al., 2013; Bian & Chen, 2017) which is widely used in nonsmooth non-Lipschitz problems.

**Definition 1.** *Let* $\psi : \mathbb{R}^d \mapsto \mathbb{R}$ *be a continuous nonsmooth, non-Lipschitz function. We call* $\tilde{\psi} : \mathbb{R}^d \times [0, +\infty] \mapsto \mathbb{R}$ *a smoothing function of* $\psi$, *if* $\tilde{\psi}(\cdot, \mu)$ *is twice continuously differentiable for any fixed* $\mu > 0$ *and* $\lim_{\hat{\boldsymbol{w}} \mapsto \boldsymbol{w}, \mu \to 0} \tilde{\psi}(\hat{\boldsymbol{w}}, \mu) = \psi(\boldsymbol{w})$ *holds for any* $\boldsymbol{w} \in \mathbb{R}^d$.

Here, we give two examples of smoothing functions. The smoothing function of $\psi_1(w) = \sum_{i=1}^{d} |w_i|$ is $\tilde{\psi}_1(w, \mu) = \sum_{i=1}^{d} (w_i^2 + \mu^2)^{1/2}$ and the smoothing function of $\psi_2(w) = \sum_{i<j} \max\{w_i, w_j\}$ is $\tilde{\psi}_2(w, \mu) = \sum_{i<j} \frac{1}{2}(\sqrt{(w_i + w_j)^2 + \mu^2} + \sqrt{(w_i - w_j)^2 + \mu^2})$.

According to Definition 1, the non-smooth lower level problem in problem (1) could be approximated by using a sequence of the following parameterized smoothing functions,

$$\boldsymbol{w}^* = \arg\min_{\boldsymbol{w}} g(\boldsymbol{w}, \bar{\boldsymbol{\lambda}}) + \exp(\lambda_1)\varphi(\tilde{h}(\boldsymbol{w}, \mu^k)) \tag{2}$$

where $\mu^k > 0$ is the smoothing parameter and $\tilde{h}(\boldsymbol{w}, \mu^k) := (\tilde{h}_1(\boldsymbol{D}_1^T\boldsymbol{w}, \mu^k), \tilde{h}_2(\boldsymbol{D}_2^T\boldsymbol{w}, \mu^k), \cdots, \tilde{h}_n(\boldsymbol{D}_n^T\boldsymbol{w}, \mu^k))$.

For each given smoothing parameter $\mu^k > 0$, we can replace the smoothed lower-level objective with its first-order necessary condition and derive the following single-level problem:

$$\min_{\boldsymbol{w}, \boldsymbol{\lambda}} f(\boldsymbol{w}, \boldsymbol{\lambda}) \quad s.t. \; c(\boldsymbol{w}, \boldsymbol{\lambda}; \mu^k) = \boldsymbol{0}, \tag{3}$$

where $c(\boldsymbol{w}, \boldsymbol{\lambda}; \mu^k) := \nabla_{\boldsymbol{w}} g(\boldsymbol{w}, \bar{\boldsymbol{\lambda}}) + \exp(\lambda_1)\nabla_{\boldsymbol{w}}\varphi(\tilde{h}(\boldsymbol{w}, \mu^k))$ and $\nabla_{\boldsymbol{w}}\varphi(\tilde{h}(\boldsymbol{w}, \mu^k)) = \varphi'(z)_{z=h(\boldsymbol{w}, \mu^k)}\nabla_{\boldsymbol{w}}h(\boldsymbol{w}, \mu^k)$.

## 3.2 STOCHASTIC CONSTRAINT GRADIENT METHOD

In this subsection, we discuss our method to solve the subproblem (3). Obviously, we can use the gradient method to solve its corresponding penalty function to solve the single-level constrained problem. However, calculating the gradient of the penalty functions needs to calculate the Hessian matrix. If the dimension of $\boldsymbol{w}$, calculating the Hessian matrix is very time-consuming. To solve this problem, we introduce a stochastic layer into the constraint such that we only need to calculate the

---

**Algorithm 1** Smoothing and Penalty Method for Non-Lipschitz Bi-level Optimization (SPNBO)

---

**Input:** $K, \mu^1, \beta^1, \delta_\mu, \delta_\epsilon \in (0,1)$.
**Output:** $\boldsymbol{w}^{k+1}$ and $\boldsymbol{\lambda}^{k+1}$.
1: **for** $k = 1, ..., K$ **do**
2:     Find $(\boldsymbol{w}^{k+1}, \boldsymbol{\lambda}^{k+1}, \boldsymbol{p}^{k+1}) := \min_{\boldsymbol{w}, \boldsymbol{\lambda}} \max_{\boldsymbol{p} \in \Delta^d} \mathcal{L}(\boldsymbol{w}^k, \boldsymbol{\lambda}^k, \boldsymbol{p}^k, \mu^k)$ using the SCG method.
3:     $\mu^{k+1} = \delta_\mu \mu^k$.
4:     $\epsilon^{k+1} = \delta_\epsilon \epsilon_k$.
5: **end for**

---

gradient of the sampled element of the constraint. Specifically, we reformulate the subproblem (3) as the following minimax problem

$$\min_{\boldsymbol{w}, \boldsymbol{\lambda}} \max_{\boldsymbol{p} \in \Delta^d} \mathcal{L}(\boldsymbol{w}, \boldsymbol{\lambda}, \boldsymbol{p}, \mu^k) = f(\boldsymbol{w}, \boldsymbol{\lambda}) + \beta \sum_{i=1}^{d} p_i c_i^2(\boldsymbol{w}, \boldsymbol{\lambda}; \mu^k) - \frac{\tau}{2} \|\boldsymbol{p}\|_2^2, \tag{4}$$

where $\beta > 0$, $\lambda > 0$, $\boldsymbol{p} \in \Delta^d := \{\boldsymbol{p} | \sum_{i=1}^{d} p_i = 1 \& 0 \le p_i \le 1\}$, $c_i(\boldsymbol{w}, \boldsymbol{\lambda}; \mu^k)$ denote the $i$-th elements of $c(\boldsymbol{w}, \boldsymbol{\lambda}; \mu^k)$. The last term is used to ensure $\mathcal{L}$ is strongly-concave on $\boldsymbol{p}$. Such a reformulation is widely used in many methods (Cotter et al., 2016; Narasimhan et al., 2020; Shi et al., 2022) to solve the constrained problem.

In each iteration, we sample an element $w_i$ of $\boldsymbol{w}$ according to distribution $p$ and calculate the corresponding value of $c_i$ and its gradient w.r.t $\boldsymbol{w}$. Then, we can obtain the stochastic gradient of $\mathcal{L}$ w.r.t $\boldsymbol{w}$ as follows,

$$\hat{\nabla}_{\boldsymbol{w}} \mathcal{L}(\boldsymbol{w}_t, \boldsymbol{\lambda}_t, \boldsymbol{p}_t, \mu^k; \xi_t) = \nabla_{\boldsymbol{w}} f(\boldsymbol{w}_t, \boldsymbol{\lambda}_t) + 2\beta c_i(\boldsymbol{w}_t, \boldsymbol{\lambda}_t; \mu^k) \nabla_{\boldsymbol{w}} c_i(\boldsymbol{w}_t, \boldsymbol{\lambda}_t; \mu^k). \tag{5}$$

Using the same method, we can obtain the stochastic gradient $\hat{\nabla}_{\boldsymbol{\lambda}} \mathcal{L}(\boldsymbol{w}_t, \boldsymbol{\lambda}_t, \boldsymbol{p}_t, \mu^k; \xi_t)$. Then,

---

**Algorithm 2** Stochastic constraint gradient (SCG)

---

**Input:** $\gamma, \sigma, \eta_t, a_{t+1,1}, a_{t+1,2}$
**Output:** $\boldsymbol{w}$ and $\boldsymbol{\lambda}$.
1: Initialize $m_{1,1}, m_{1,2}, m_{1,3}, \hat{m}_{t,1}, \hat{m}_{t,2}, \hat{m}_{t,3}, \eta_1$.
2: **while** Not satisfy the conditions (10) **do**
3:     $\tilde{\boldsymbol{w}}_{t+1} = \boldsymbol{w}_t - \gamma A_{t,1}^{-1} m_{t,1}$.
4:     $\boldsymbol{w}_{t+1} = \boldsymbol{w}_t + \eta_t(\tilde{\boldsymbol{w}}_{t+1} - \boldsymbol{w}_t)$.
5:     $\tilde{\boldsymbol{\lambda}}_{t+1} = \boldsymbol{\lambda}_t - \gamma A_{t,2}^{-1} m_{t,2}$.
6:     $\boldsymbol{\lambda}_{t+1} = \boldsymbol{\lambda}_t + \eta_t(\tilde{\boldsymbol{\lambda}}_{t+1} - \boldsymbol{\lambda}_t)$.
7:     $\tilde{\boldsymbol{p}}_{t+1} = \mathcal{P}_\Delta(\boldsymbol{p}_t + \sigma A_{t,3}^{-1} m_{t,3})$.
8:     $\boldsymbol{p}_{t+1} = \boldsymbol{p}_t + \eta_t(\tilde{\boldsymbol{p}}_{t+1} - \boldsymbol{p}_t)$.
9:     Sample a constraint according to distribution $\boldsymbol{p}$.
10:     Calculate the stochastic gradient $\hat{\nabla}_{\boldsymbol{w}} \mathcal{L}(\boldsymbol{w}_t, \boldsymbol{\lambda}_t, \boldsymbol{p}_t, \mu^k; \xi_t)$ and $\hat{\nabla}_{\boldsymbol{\lambda}} \mathcal{L}(\boldsymbol{w}_t, \boldsymbol{\lambda}_t, \boldsymbol{p}_t, \mu^k; \xi_t)$.
11:     Randomly sample a constraint.
12:     Calculate the stochastic gradient $\hat{\nabla}_{\boldsymbol{p}} \mathcal{L}(\boldsymbol{w}_t, \boldsymbol{\lambda}_t, \boldsymbol{p}_t, \mu^k; \xi_t)$.
13:     Update $m_{t,1}, m_{t,2}, m_{t,3}$.
14:     Update $\hat{m}_{t,1}, \hat{m}_{t,2}, \hat{m}_{t,3}$ and clip to $[\rho, b]$.
15:     Calculate $A_{t,1}, A_{t,2}, A_{t,3}$.
16: **end while**

---

randomly sample another element $w_j$, we can calculate the value of $c_j$ and obtain the stochastic gradient of $\mathcal{L}$ w.r.t. $\boldsymbol{p}$ as follows

$$\hat{\nabla}_{\boldsymbol{p}} \mathcal{L}(\boldsymbol{w}_t, \boldsymbol{\lambda}_t, \boldsymbol{p}_t, \mu^k; \xi_t) = de_j(c_j^2(\boldsymbol{w}_t, \boldsymbol{\lambda}_t; \mu^k) - \lambda p_j), \tag{6}$$

where $e_j$ denotes a vector where its $j$-th element is 1 and other elements are 0. Since $\boldsymbol{p}$ is related to the value of constraints, sampling constraint according to $\boldsymbol{p}$ helps us find the most violating conditions.

To achieve a better performance, the momentum-based variance reduction method and adaptive method are also used. Specifically, we calculate the momentum-based gradient estimation w.r.t $\boldsymbol{w}, \boldsymbol{\lambda}$

and $\boldsymbol{p}$ as follows,

$$m_{t,1} = \hat{\nabla}_{\boldsymbol{w}}\mathcal{L}(\boldsymbol{w}_t, \boldsymbol{\lambda}_t, \boldsymbol{p}_t, \mu^k; \xi_t) + (1 - a_{t+1,1})\left(m_{t-1,1} - \hat{\nabla}_{\boldsymbol{w}}\mathcal{L}(\boldsymbol{w}_{t-1}, \boldsymbol{\lambda}_{t-1}, \boldsymbol{p}_{t-1}, \mu^k; \xi_t)\right)$$

$$m_{t,2} = \hat{\nabla}_{\boldsymbol{\lambda}}\mathcal{L}(\boldsymbol{w}_t, \boldsymbol{\lambda}_t, \boldsymbol{p}_t, \mu^k; \xi_t) + (1 - a_{t+1,1})\left(m_{t-1,2} - \hat{\nabla}_{\boldsymbol{\lambda}}\mathcal{L}(\boldsymbol{w}_{t-1}, \boldsymbol{\lambda}_{t-1}, \boldsymbol{p}_{t-1}, \mu^k; \xi_t)\right)$$

$$m_{t,3} = \hat{\nabla}_{\boldsymbol{p}}\mathcal{L}(\boldsymbol{w}_t, \boldsymbol{\lambda}_t, \boldsymbol{p}_t, \mu^k; \xi_t) + (1 - a_{t+1,2})\left(m_{t-1,3} - \hat{\nabla}_{\boldsymbol{p}}\mathcal{L}(\boldsymbol{w}_{t-1}, \boldsymbol{\lambda}_{t-1}, \boldsymbol{p}_{t-1}, \mu^k; \xi_t)\right)$$

Then, we calculate the adaptive matrix matrices $A_{t,1}$, $A_{t,2}$ and $A_{t,3}$ for updating $\boldsymbol{w}$, $\boldsymbol{\lambda}$ and $\boldsymbol{p}$, respectively. Here, we present the calculation of adaptive matrix $A_{t,1}$ as an example. Specifically, we calculate a second momentum-based estimation $\hat{m}_{t,1} = \hat{a}\hat{\nabla}_{\boldsymbol{w}}\mathcal{L}(\boldsymbol{w}_t, \boldsymbol{\lambda}_t, \boldsymbol{p}_t, \mu^k; \xi_t)^2 + (1 - \hat{a})m_{t-1,1}$. Then, we clip each element of $\hat{m}_{t,1}$ into the range of $[\rho, b]$ and obtain the adaptive matrix $A_{t,1} = diag\left(\sqrt{clip(\hat{m}_{t,1}, \rho, b)}\right)$. Note that we can use other method to calculate the adaptive matrices, such as AdaGrad-Norm (Ward et al., 2020), AMSGrad (Reddi et al., 2019), Adam$^+$ (Liu et al., 2020). Then, we can obtain the update rules as follows,

$$\tilde{\boldsymbol{w}}_{t+1} = \boldsymbol{w}_t - \gamma A_{t,1}^{-1}m_{t,1}, \quad \boldsymbol{w}_{t+1} = \boldsymbol{w}_t + \eta_t(\tilde{\boldsymbol{w}}_{t+1} - \boldsymbol{w}_t), \tag{7}$$

$$\tilde{\boldsymbol{\lambda}}_{t+1} = \boldsymbol{\lambda}_t - \gamma A_{t,2}^{-1}m_{t,2}, \quad \boldsymbol{\lambda}_{t+1} = \boldsymbol{\lambda}_t + \eta_t(\tilde{\boldsymbol{\lambda}}_{t+1} - \boldsymbol{\lambda}_t), \tag{8}$$

$$\tilde{\boldsymbol{p}}_{t+1} = \mathcal{P}_\Delta\left(\boldsymbol{p}_t + \sigma A_{t,3}^{-1}m_{t,3}\right), \quad \boldsymbol{p}_{t+1} = \boldsymbol{p}_t + \eta_t(\tilde{\boldsymbol{p}}_{t+1} - \boldsymbol{p}_t), \tag{9}$$

where $\gamma > 0$, $\sigma > 0$, $\eta_t > 0$ and $\mathcal{P}_\Delta(\cdot)$ denotes the projection onto $\Delta^d$.

Once the following conditions are satisfied,

$$\|\nabla_{\boldsymbol{w}}\mathcal{L}(\boldsymbol{w}, \boldsymbol{\lambda}, \boldsymbol{p}, \mu^k)\|_2^2 \leq \epsilon_k^2, \quad \|\nabla_{\boldsymbol{\lambda}}\mathcal{L}(\boldsymbol{w}, \boldsymbol{\lambda}, \boldsymbol{p}, \mu^k)\|_2^2 \leq \epsilon_k^2, \quad \|c(\boldsymbol{w}, \boldsymbol{\lambda}; \mu^k)\|_2^2 \leq \epsilon_k^2, \tag{10}$$

where $\nabla_{\boldsymbol{w}}\mathcal{L}$ and $\nabla_{\boldsymbol{\lambda}}\mathcal{L}$ denote the full gradients $\mathcal{L}$ w.r.t $\boldsymbol{w}$ and $\boldsymbol{\lambda}$, we enlarge the penalty parameter $\beta$, and decrease the smooth parameter $\mu^k$.

The whole algorithm is presented in Algorithm 1 and 2. Note that instead of checking the conditions in each iteration of SCD, we check the conditions after several iterations to save time.

## 4    THEORETICAL ANALYSIS

In this section, we discuss the convergence performance of our proposed method (Detailed proofs are given in our appendix). Here, we give several assumptions which are widely used in convergence analysis.

**Assumption 1.** *We have* $\mathbb{E}[\hat{\nabla}_{\boldsymbol{z}}\mathcal{L}(\boldsymbol{z}_{t+1}, \boldsymbol{p}_{t+1}, \mu^k)] = \nabla_{\boldsymbol{z}}\mathcal{L}(\boldsymbol{z}_{t+1}, \boldsymbol{p}_{t+1}, \mu^k), \mathbb{E}[\hat{\nabla}_{\boldsymbol{p}}\mathcal{L}(\boldsymbol{z}_{t+1}, \boldsymbol{p}_{t+1}, \mu^k))] = \nabla_{\boldsymbol{p}}\mathcal{L}(\boldsymbol{z}_{t+1}, \boldsymbol{p}_{t+1}, \mu^k)$, $\mathbb{E}[\|\hat{\nabla}_{\boldsymbol{z}}\mathcal{L}(\boldsymbol{z}_{t+1}, \boldsymbol{p}_{t+1}, \mu^k) - \nabla_{\boldsymbol{z}}\mathcal{L}(\boldsymbol{z}_{t+1}, \boldsymbol{p}_{t+1}, \mu^k)\|_2] \leq \mathbb{V}_z^2$ *and* $\mathbb{E}[\|\hat{\nabla}_{\boldsymbol{p}}\mathcal{L}(\boldsymbol{z}_{t+1}, \boldsymbol{p}_{t+1}, \mu^k)) - \nabla_{\boldsymbol{p}}\mathcal{L}(\boldsymbol{z}_{t+1}, \boldsymbol{p}_{t+1}, \mu^k))\|_2] \leq \mathbb{V}_p^2.$.

**Assumption 2.** *The function* $\mathcal{L}(\boldsymbol{w}, \boldsymbol{\lambda}, \boldsymbol{p}, \mu^k)$ *is* $\tau$*-strongly concave on* $\boldsymbol{p}$ *for any give* $\mu^k > 0$.

**Assumption 3.** *The function* $\mathcal{L}(\boldsymbol{w}, \boldsymbol{\lambda}, \boldsymbol{p}, \mu^k)$ *has a* $L_1$*-Lipschitz gradient on* $(\boldsymbol{w}, \boldsymbol{\lambda}, \boldsymbol{p})$ *for any give* $\mu^k > 0$.

**Assumption 4.** *The smoothing function* $\tilde{h}(\boldsymbol{w}, \mu^k)$ *is twice continuously differentiable on* $\boldsymbol{w}$ *for any* $\mu^k > 0$.

**Assumption 5.** $f$ *is Lipschitz continuous and* $g$ *is twice Lipschitz continuous w.r.t.* $\boldsymbol{w}$ *and* $\boldsymbol{\lambda}$.

We prove our Algorithm 2 can converge to the points satisfying conditions 10. Here, we give the definitions of $\epsilon$-stationary of the constrained problem 3 and minimax problem 4 and then show the relations between these definitions as follows,

**Definition 2.** *(*$\epsilon$*-stationary point of the constrained optimization problem.)* $(\boldsymbol{w}^*, \boldsymbol{\lambda}^*, \boldsymbol{\alpha}^*)$ *is said to be the* $\epsilon$*-stationary point of the sub-problem (3) if the following conditions hold,* $\|\nabla_{\boldsymbol{w}}f(\boldsymbol{w}^*, \boldsymbol{\lambda}^*; \mu^k) + \sum_{i=1}^d \alpha_i^*\nabla_{\boldsymbol{w}}c_i(\boldsymbol{w}^*, \boldsymbol{\lambda}^*)\|_2^2 \leq \epsilon_1^2$, $\|\nabla_{\boldsymbol{\lambda}}f(\boldsymbol{w}^*, \boldsymbol{\lambda}^*) + \sum_{i=1}^d \alpha_i^*\nabla_{\boldsymbol{\lambda}}c_i(\boldsymbol{w}^*, \boldsymbol{\lambda}^*; \mu^k)\|_2^2 \leq \epsilon_2^2$ *and* $\sum_{i=1}^d c_i^2(\boldsymbol{w}^*, \boldsymbol{\lambda}^*; \mu^k) \leq \epsilon_3^2$, *where* $\boldsymbol{\alpha}$ *denotes the lagrangian multipliers.*

**Remark 1.** *Let* $\alpha_j^* = p_j^*2c_j(\boldsymbol{w}^*, \boldsymbol{\lambda}^*; \mu^k)$. *The conditions in Defintion 2 is equivalent to the tolerance conditions 10.*

**Definition 3.** (*$\epsilon$-stationary point of the mini-max problem.*) $(\boldsymbol{w}^*, \boldsymbol{\lambda}^*, \boldsymbol{p}^*)$ *is said to be the $\epsilon$-stationary point of the mini-max problem if it satisfies the conditions* $\|\nabla_{\boldsymbol{w}}\mathcal{L}\|_2^2 \le \epsilon^2$, $\|\nabla_{\boldsymbol{\lambda}}\mathcal{L}\|_2^2 \le \epsilon^2$ *and* $\|\nabla_{\boldsymbol{p}}\mathcal{L}\|_2^2 \le \epsilon^2$.

**Proposition 1.** *If Assumptions 2 and 3 hold, $(\boldsymbol{w}^*, \boldsymbol{\lambda}, \boldsymbol{p}^*)$ is the $\epsilon$-stationary point of the problem (4), then $(\boldsymbol{w}^*, \boldsymbol{\lambda}^*)$ is the $\epsilon$-stationary point of the constrained problem 3.*

According to Shi et al. (2022), the minimax problem 4 is equivalent to the following minimization problem:

$$\min_{\boldsymbol{w}, \boldsymbol{\lambda}} \left\{ H(\boldsymbol{w}, \boldsymbol{\lambda}) := \max_{\boldsymbol{p} \in \Delta^m} \mathcal{L}(\boldsymbol{w}, \boldsymbol{\lambda}, \boldsymbol{p}) = \mathcal{L}(\boldsymbol{w}, \boldsymbol{\lambda}, \boldsymbol{p}^*(\boldsymbol{w}, \boldsymbol{\lambda})) \right\}, \tag{11}$$

where $\boldsymbol{p}^*(\boldsymbol{w}, \boldsymbol{\lambda}) = \arg\max_{\boldsymbol{p}} \mathcal{L}(\boldsymbol{w}, \boldsymbol{\lambda}, \boldsymbol{p})$. Here, we give stationary point the minimization problem 11 and its relationship with Definition 3 as follows,

**Definition 4.** *We call $\boldsymbol{w}^*$ an $\epsilon$-stationary point of a differentiable function $H(\boldsymbol{w})$, if $\|\nabla_{\boldsymbol{w}}H(\boldsymbol{w}^*, \boldsymbol{\lambda}^*)\|_2 \le \epsilon$ and $\|\nabla_{\boldsymbol{\lambda}}H(\boldsymbol{w}^*, \boldsymbol{\lambda}^*)\|_2 \le \epsilon$.*

**Proposition 2.** *Under Assumptions 3 and 2, if $(\boldsymbol{w}', \boldsymbol{\lambda}')$ is the $\epsilon$-stationary point of $H(\boldsymbol{w}, \boldsymbol{\lambda})$, then $(\boldsymbol{w}', \boldsymbol{\lambda}', \boldsymbol{p}')$ is the $\epsilon$-stationary point of $\min_{\boldsymbol{w}, \boldsymbol{\lambda}} \max_{\boldsymbol{p} \in \Delta^d} \mathcal{L}(\boldsymbol{w}, \boldsymbol{\lambda}, \boldsymbol{p}, \mu^k)$ can be obtained. Conversely, if $(\boldsymbol{w}', \boldsymbol{\lambda}', \boldsymbol{p}')$ is the $\epsilon$-stationary point of $\min_{\boldsymbol{w}, \boldsymbol{\lambda}} \max_{\boldsymbol{p} \in \Delta^d} \mathcal{L}(\boldsymbol{w}, \boldsymbol{\lambda}, \boldsymbol{p}, \mu^k)$, then a point $(\boldsymbol{w}', \boldsymbol{\lambda}')$ is stationary point of $H(\boldsymbol{w}, \boldsymbol{\lambda})$.*

**Remark 2.** *According to Proposition 1 and Proposition 2, we have that once we find the $\epsilon$-stationary point in terms of Definition 4, then we can get the $\epsilon$-stationary point in terms of Definition 2. Therefore, we can obtain the points satisfying the tolerance conditions (10).*

Before, we give the convergence reuslt of our method, we present the lemma useful in our analysis. We have

**Lemma 1.** *Under assumptions, let $\boldsymbol{z} = [\boldsymbol{w}; \boldsymbol{\lambda}]$, we have*

$$\|\nabla H(\boldsymbol{z}) - m_{t,z}\|_2^2 \le 2L_1^2 \|\boldsymbol{p}^*(\boldsymbol{z}_t) - \boldsymbol{p}_t\|_2^2 + 2\|\nabla_{\boldsymbol{z}}\mathcal{L}(\boldsymbol{z}_t, \boldsymbol{p}_t, \mu^k) - m_{t,z}\|_2^2 \tag{12}$$

Then, we can define the following metric

$$\mathcal{M}_k = \frac{b^2}{\gamma^2} \|\tilde{\boldsymbol{z}}_{t+1} - \boldsymbol{z}_t\|_2^2 + 2L_1^2 \|\boldsymbol{p}^*(\boldsymbol{z}_t) - \boldsymbol{p}_t\|_2^2 + 2\|\nabla_{\boldsymbol{z}}\mathcal{L}(\boldsymbol{z}_t, \boldsymbol{p}_t, \mu^k) - m_{t,z}\|_2^2. \tag{13}$$

We have

$$\mathcal{M}_k \ge \frac{b^2}{\gamma^2} \|\boldsymbol{z}_t - \gamma A_{t,z}^{-1} m_{t,z} - \boldsymbol{z}_t\|_2^2 + \|\nabla H(\boldsymbol{z}) - m_{t,z}\|_2^2 \ge \|m_{t,z}\|_2^2 + \|\nabla H(\boldsymbol{z}) - m_{t,z}\|_2^2 \ge \frac{1}{2}\nabla H(\boldsymbol{z}_t)$$

If $\mathcal{M}_k \to 0$, we have $\|\nabla H(\boldsymbol{z}_t)\|_2^2 \to 0$. Thus, we can bound $\mathcal{M}_k$ to find the stationary point of problem (11). Then, we give the convergence theorem in the following theorem.

**Theorem 1.** *Assume Assumptions hold, if $a_{t+1,1} = c_1 \eta_t^2, a_{t+1,2} = c_2 \eta_t^2, c_1 > \frac{5}{2} + \frac{2}{3e^3}\eta_t, 0 < \gamma \le$*

$$\frac{\sqrt{3}\tau\sigma\rho}{2\sqrt{12L_1^2\sigma^2\kappa^2 + 125L_1^2\kappa^2 b^2}} \text{ and } 0 < \sigma \le \min\{\frac{15}{12\mu}, \frac{1}{6L_1}\}, \text{ we have}$$

$$\frac{\gamma}{4\rho}\frac{1}{T}\sum_{t=1}^{T}\mathcal{M}_t \le \frac{(\Theta_1 - \Theta^*)(m+T)^{1/3}}{Te} + \frac{\gamma(c_1^2\mathbb{V}_z^2 + c_2^2\mathbb{V}_p^2)(m+T)^{1/3}}{\rho T}\ln(m+T) \tag{14}$$

**Remark 3.** *Theorem 1 demonstrate that with suitable setting, our method can converge to the points satisfying the conditions (10) at the rate of $\tilde{\mathcal{O}}(T^{-2/3})$ if omitting $\log$.*

Then, we discuss the convergence performance of our whole algorithm. Define a new function

$$h_i^{\bar{\boldsymbol{w}}}(\boldsymbol{D}_i^T \boldsymbol{w}) := \begin{cases} h_i(\boldsymbol{D}_i^T \boldsymbol{w}) & i \notin \mathcal{I}_{\bar{\boldsymbol{w}}} \\ h_i(\boldsymbol{D}_i^T \bar{\boldsymbol{w}}) & i \in \mathcal{I}_{\bar{\boldsymbol{w}}} \end{cases}, \tag{15}$$

which is Lipschitz continuous at $\boldsymbol{D}_i^T \bar{\boldsymbol{w}}$, $i = 1, 2, \cdots, n$. Then, we have $h_{\bar{\boldsymbol{w}}}(\boldsymbol{w}) := (h_1^{\bar{\boldsymbol{w}}}(\boldsymbol{D}_1^T \boldsymbol{w}), h_2^{\bar{\boldsymbol{w}}}(\boldsymbol{D}_2^T \boldsymbol{w}), \cdots, h_n^{\bar{\boldsymbol{w}}}(\boldsymbol{D}_n^T \boldsymbol{w}))$, which has the same value as $h(\boldsymbol{w})$ but opposite property.

For convenience, we define $\phi_{\bar{\boldsymbol{w}}}(\boldsymbol{w}) = \varphi(h_{\bar{\boldsymbol{w}}}(\boldsymbol{w}))$ and $\phi(\boldsymbol{w}) = \varphi(h(\boldsymbol{w}))$. Besides, we define a vector set as follows,

$$\mathcal{V}_{\bar{\boldsymbol{w}}} = \left\{ \boldsymbol{v} : \boldsymbol{D}_i^T \boldsymbol{v} = \boldsymbol{0}, i \in \mathcal{I}_{\bar{\boldsymbol{w}}} \right\}, \tag{16}$$

which means that $\boldsymbol{v}$ is perpendicular to all column vectors in $\boldsymbol{D}_i, i \in \mathcal{I}_{\bar{\boldsymbol{w}}}$. According to Bian & Chen (2017), the necessary condition of the non-Lipschitz lower level problem is

$$\nabla_{\boldsymbol{w}} g(\boldsymbol{w}^*, \bar{\boldsymbol{\lambda}})^T \boldsymbol{v} + \exp(\lambda_1)\phi^{\circ}(\boldsymbol{w}^*; \boldsymbol{v}) \geq 0, \tag{17}$$

for all $\boldsymbol{v} \in \mathcal{V}_{\boldsymbol{w}^*}$, where $\phi^{\circ}(\boldsymbol{w}^*; \boldsymbol{v}) = \limsup\limits_{\substack{\boldsymbol{w} \mapsto \boldsymbol{w}^* \\ t \downarrow 0}} \dfrac{\phi(\boldsymbol{w} + t\boldsymbol{v}) - \phi(\boldsymbol{w})}{t}$ denotes the Clarke generalized directional derivative of $\varphi(h(\boldsymbol{w}))$ at $\boldsymbol{w}^*$. Replacing the lower-level problem with above condition, we can obtain the following single-level problem,

$$\min_{\boldsymbol{w}, \boldsymbol{\lambda}} f(\boldsymbol{w}, \boldsymbol{\lambda}) \quad s.t. \quad c(\boldsymbol{w}, \boldsymbol{\lambda}) = \nabla_{\boldsymbol{w}} g(\boldsymbol{w}, \bar{\boldsymbol{\lambda}})^T \boldsymbol{v} + \exp(\lambda_1)\phi^{\circ}(\boldsymbol{w}; \boldsymbol{v}) \geq 0 \tag{18}$$

for all $\boldsymbol{v} \in \mathcal{V}_{\boldsymbol{w}^*}$. For this new problem, we have the following theorem.

**Theorem 2.** *If $(\boldsymbol{w}^*, \boldsymbol{\lambda}^*)$ satisfy the following conditions, then they are the stationary points of the problem (18).*

$$\nabla_{\boldsymbol{w}} f(\boldsymbol{w}^*, \boldsymbol{\lambda}^*)^T \boldsymbol{v}_2 - (\boldsymbol{v}_2^T \nabla_{\boldsymbol{w}\boldsymbol{w}}^2 g(\boldsymbol{w}^*, \bar{\boldsymbol{\lambda}}^*)\boldsymbol{v}_1 + \exp(\lambda_1^*)\phi^{\circ\circ}(\boldsymbol{w}^*; \boldsymbol{v}_1, \boldsymbol{v}_2))\xi^* \geq 0, \tag{19}$$

$$\nabla_{\boldsymbol{\lambda}} f(\boldsymbol{w}^*, \boldsymbol{\lambda}^*)^T \boldsymbol{v}_3 - (\bar{\boldsymbol{v}}_3 \nabla_{\boldsymbol{w}\bar{\boldsymbol{\lambda}}}^2 g(\boldsymbol{w}^*, \bar{\boldsymbol{\lambda}}^*)\boldsymbol{v}_1 + v_3^1 \exp(\lambda_1^*)\phi^{\circ}(\boldsymbol{w}^*; \boldsymbol{v}_1))\xi^* \geq 0, \tag{20}$$

$$\nabla_{\boldsymbol{w}} g(\boldsymbol{w}^*, \bar{\boldsymbol{\lambda}}^*)^T \boldsymbol{v}_1 + \exp(\lambda_1^*)\phi^{\circ}(\boldsymbol{w}^*; \boldsymbol{v}_1) \geq 0, \tag{21}$$

$$\xi^* \left( \nabla_{\boldsymbol{w}} g(\boldsymbol{w}^*, \bar{\boldsymbol{\lambda}}^*)^T \boldsymbol{v}_1 + \lambda_1^* \phi^{\circ}(\boldsymbol{w}^*; \boldsymbol{v}_1) \right) = 0, \tag{22}$$

$$\xi^* \geq 0, \tag{23}$$

*for all $\boldsymbol{v}_1 \in \mathcal{V}_{\boldsymbol{w}^*}$, $\boldsymbol{v}_2 \in \mathbb{R}^d$, $\boldsymbol{v}_3 \in \mathbb{R}^m$, where $\boldsymbol{v}_3 = [v_3^1, \bar{\boldsymbol{v}}_3^T]^T$, $\bar{\boldsymbol{v}}_3^T = [v_3^2, \cdots, v_3^m]^T$ and $\phi^{\circ\circ}(\boldsymbol{w}^*; \boldsymbol{v}_1, \boldsymbol{v}_2) = \limsup\limits_{\substack{\boldsymbol{w} \mapsto \boldsymbol{w}^*, \\ s \downarrow 0}} \dfrac{\phi^{\circ}(\boldsymbol{w} + \boldsymbol{v}_2 s; \boldsymbol{v}_1) - \phi^{\circ}(\boldsymbol{w}; \boldsymbol{v}_1)}{s}$). In addition, $(\boldsymbol{v}_1, \boldsymbol{v}_2, \boldsymbol{v}_3)$ is direction vector used in calculating the Clarke directional derivative.*

Then, we show with decreasing the smoothing parameter and tolerance parameters, our method can converge to stationary point defined in Theorem 2 in the following theorem.

**Theorem 3.** *Suppose $\{\epsilon_k\}_{k=1}^{\infty}$ are positive and convergent ($\lim_{k \to \infty} \epsilon_k = 0$) sequences, $\{\mu^k\}_{k=1}^{\infty}$ is a positive and convergent ($\lim_{k \to \infty} \mu^k = 0$) sequence. Then any limit point of the sequence points generated by SPNBO satisfies the conditions (19)-(23).*

Then, we show the relations between the conditions 19-23 and the original nonsmooth bilevel problem (1).

**Theorem 4.** *Assume the lower level problem in problem (1) is strongly convex. If we have $(\boldsymbol{w}^*, \boldsymbol{\lambda}^*)$ and $\xi^* \geq 0$ satisfying the conditions (19)-(23), then $(\boldsymbol{w}^*, \boldsymbol{\lambda}^*)$ is the stationary point of the original nonsmooth bilevel problem.*

## 5 EXPERIMENTS

In this section, we conduct experiments to demonstrate the superiority of our method in terms of accuracy and efficiency.

### 5.1 EXPERIMENTAL SETUP

We summarize the baseline methods used in our experiments as follows.

1. **Penalty**. The method proposed in Mehra & Hamm (2019). It formulates the bi-level optimization problem as a one-level optimization problem, and then use penalty method to solve the new problem.

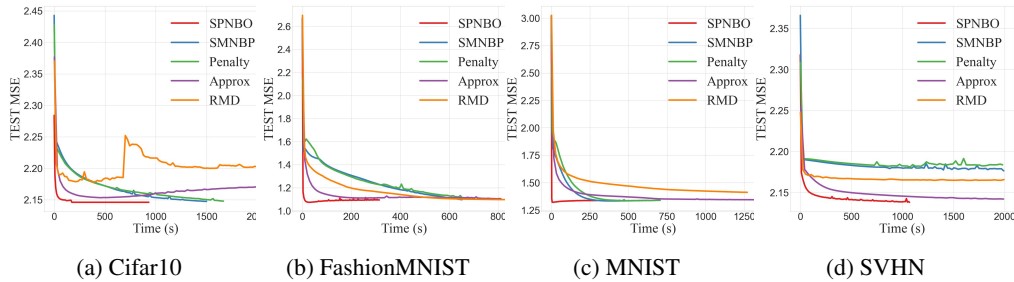

|   (a) Cifar10   |   (b) FashionMNIST   |   (c) MNIST   |   (d) SVHN   |

Figure 1: Test MSE against training time of all the methods in data re-weight.

2. **Approx**. The method proposed in Pedregosa (2016). It solves an additional linear problem to find the hypergradient to update the hyper-parameters.

3. **RMD**. The reverse method proposed in Franceschi et al. (2017). An additional loop is used to approximate the hypergradient.

4. **SMNBP**. The method proposed in Okuno et al. (2021). It uses the smoothing method to produce a sequence of smoothing lower-level functions and replaces them with the necessary condition. Then the penalty method is used to solve each single level problem.

We implement SMNBP, Penalty, Approx, RMD, and our method in Python. Since original Penalty, Approx and RMD are used for the smoothing problems, we use the smoothing function to approximate the lower-level problem. We fix the smoothing parameter at $\mu = 0.0001$ in these methods. In SMNBP, for each given smoothing parameter, we solve the constrained problem using the Penalty method. For all these method, we use ADAM to update $\boldsymbol{w}$ and $\boldsymbol{\lambda}$ and choose the initial step size from $\{0.1, 0.01, 0.001\}$. For our method, we set $\hat{a} = 0.9$ and other parameters are set according to Theorem 1. We choose the penalty parameter from $\{0.1, 1, 10, 100\}$ for our method, SMNBP, and Penalty. We fix the inner iteration number $T$ in Penalty, Approx, and RMD at 10 according to Mehra & Hamm (2019). We summarize the datasets used in our experiments in Table 2 and we divide all the datasets into three parts, *i.e.*, 40% for training, 40% for validation and 20% for testing. All the experiments are carried out 10 times on a PC with four 1080 Ti GPUs.

## 5.2 APPLICATIONS

**Data re-weight:** In this experiment, we evaluate the performance of all the methods in the application named data re-weight. In many real-world applications, the training set and testing set may have different distributions. To reduce the discrepancy between the two distributions, each data point will be given an additional importance weight, which is called data re-weight. In this application, we search the weight $\lambda_i$ of each training data and the OSCAR regularization parameters $\hat{\lambda}$ and $\check{\lambda}$. This problem can be formulated as

Table 2: Datasets used in the experiments.

| Datasets | Features | Samples | Classes |
|---|---|---|---|
| SVHN | $32 \times 32 \times 3$ | 73257 | 10 |
| Cifar10 | $84 \times 84 \times 3$ | 50000 | 10 |
| MNIST | $28 \times 28 \times 1$ | 60000 | 10 |
| FashionMNIST | $28 \times 28 \times 1$ | 60000 | 10 |

$$\min_{\boldsymbol{\lambda}} \frac{1}{N_{val}} \sum_{i=1}^{N_{val}} \ell(\boldsymbol{x}_i^T \boldsymbol{w}^*, \boldsymbol{y}_i) \tag{24}$$

$$s.t. \ \boldsymbol{w}^* \in \arg\min_{\boldsymbol{w}} \sum_{i=1}^{N_{tr}} \frac{\exp(\lambda_i)}{\sum_j \exp(\lambda_j)} \ell(\boldsymbol{x}_i^T \boldsymbol{w}, \boldsymbol{y}_i) + \exp(\hat{\lambda}) \|\boldsymbol{w}\|_1 + \exp(\check{\lambda}) \sum_{i<j} \max\{\boldsymbol{w}_{\mathcal{G}_i}, \boldsymbol{w}_{\mathcal{G}_j}\},$$

where $N_{tr}$ and $N_{val}$ denote the sizes of training set and validation set respectively. $\{\boldsymbol{x}_i, \boldsymbol{y}_i\}$ denotes the data instance, $\mathcal{G}_i$ denotes the group index. In this experiments, we set the group number equal to 10, and we use the mean squared loss $\ell = (\boldsymbol{x}_i^T \boldsymbol{w} - \boldsymbol{y}_i)^2$.

**Training data poisoning:** In this experiment, we evaluate the performance of all the methods in training data poisoning. Assume we have pure training data $\{\boldsymbol{x}_i\}_{i=1}^{N_{tr}}$ with several poisoned points

Table 3: The test mse of all the methods in data reweight. (Lower is better.)

| Datasets | SPNBO | SMNBP | Penalty | Approx | RMD |
|---|---|---|---|---|---|
| Cifar10 | **2.146** ± 0.006 | 2.147 ± 0.012 | 2.147 ± 0.011 | 2.171 ± 0.004 | 2.203 ± 0.012 |
| MNIST | **1.338** ± 0.004 | 1.339 ± 0.006 | 1.340 ± 0.007 | 1.345 ± 0.010 | 1.412 ± 0.076 |
| FashionMNIST | **1.091** ± 0.011 | 1.096 ± 0.020 | 1.100 ± 0.001 | 1.104 ± 0.009 | 1.097 ± 0.013 |
| SVHN | **2.138** ± 0.004 | 2.176 ± 0.002 | 2.184 ± 0.006 | 2.142 ± 0.004 | 2.165 ± 0.002 |

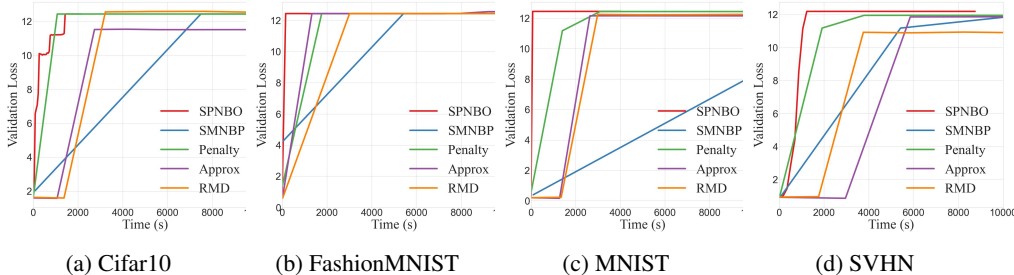

(a) Cifar10      (b) FashionMNIST      (c) MNIST      (d) SVHN

Figure 2: Validation loss versus training time of all the methods in training data poisoning.

$\{\boldsymbol{\lambda}_j\}_{j=1}^{N_{poi}}$ assigned arbitrary labels. In this task, we search the poisoned data which can hurt the performance of the model trained from the clean data. This problem can be formulated as

$$\max_{\boldsymbol{\lambda}} \frac{1}{N_{val}} \sum_{i=1}^{N_{val}} \ell(\theta(\boldsymbol{x}_i; \boldsymbol{w}^*), \boldsymbol{y}_i) \quad s.t. \quad \boldsymbol{w}^* \in \arg\min_{\boldsymbol{w}} \frac{1}{N} \sum_{\boldsymbol{x}_i \in \mathcal{D}} \ell(\theta(\boldsymbol{x}_i; \boldsymbol{w}), \boldsymbol{y}_i) + \|\boldsymbol{w}\|_p^p,$$

where $N = N_{tr} + N_{poi}$ and $\mathcal{D}$ denote the dataset containing all the clean training data and poisoned data. In this experiment, we use Resnet18 as model. Besides, we add a $p$-norm ($0 < p < 1$) regularization term in the lower-level problem to ensure that we can get a sparse model. In this experiment, we set $p = 0.5$. After solving the bilevel problem, we retrain the model on the clean data and poisoned data and then test the model.

Table 4: Test accuracy (%) of all the methods in training data poisoning (lower is better).

| Data | Approx | RMD | Penalty | SMNBP | SPNBO |
|---|---|---|---|---|---|
| SVHN | 50.79 ± 0.39 | 50.67 ± 0.27 | 50.67 ± 0.27 | 50.62 ± 0.29 | **48.85** ± 0.57 |
| Cifar10 | 82.91 ± 0.18 | 83.25 ± 0.11 | 82.29 ± 0.11 | 82.57 ± 0.11 | **82.22** ± 0.28 |
| FashionMNIST | 96.09 ± 0.07 | 95.89 ± 0.31 | 95.87 ± 0.19 | 96.01 ± 0.22 | **95.80** ± 0.20 |
| MNIST | 80.27 ± 0.25 | 77.63 ± 0.08 | 77.43 ± 0.54 | 77.50 ± 0.30 | **77.22** ± 0.08 |

### 5.3 RESULTS AND DISCUSSION

All the results are presented in Tables 3, 4 and Figure 1, 2. From Table 3 and Table 4, we can find that our method has the similar results to other methods. From Figure 1 and Figure 2, we can find that our method is faster than other methods in most cases. This is because Approx and RMD need to solve the lower-level objective first and then need an additional loop to approximate the hypergradient which makes these methods have higher time complexity. Penalty and SMNBP need to use all the constraints in each updating step which is also time-consuming, when we use complex models (*e.g.*, DNNs), Penalty and SMNBP suffer from high time complexity. However, our method uses the stochastic gradient method which makes it scalable to complicated models and does not need any intermediate steps to approximate the hypergradient. From all these results, we can conclude that our SPNBO is superior to other methods in terms of accuracy and efficiency.

## 6 CONCLUSION

In this paper, we proposed a new method, SPNBO, to solve the generalized non-smooth non-Lipschitz bi-level optimization problems by using the smoothing method and the penalty method. We also give the convergence analysis of our proposed method. The experimental results demonstrate the superiority of our method in terms of training time and accuracy.

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
