# OpenReview forum: "Efficient Method for Bi-level Optimization with Non-smooth Lower-Level Problem"
_ICLR.cc/2023/Conference — Submitted to ICLR 2023_

### Official Review · Reviewer_xEMd · 2022-10-21

**Confidence:** 4
**Correctness:** 3
**Technical Novelty And Significance:** 2
**Empirical Novelty And Significance:** 2
**Recommendation:** 3

**Clarity, Quality, Novelty And Reproducibility:**

In addition to be a rather limited novelty, the smoothing approach seems to introduce a lot a new hyperparameters controlling the convergence of the algorithm, in particular hyperparameters controlling the decrease of the smoothing.


**Strength And Weaknesses:**

My main concerns are the following:

- Could you clarify the contribution, is it the smoothing? Or the stochasticity in the algorithm?

- "Instead of calculating the hypergradients which normally involves the calculation of Hessian matrices or the training of lower-level problem"
Can authors give a reference for the fact that single-level problem requires creating the Hessian  I am very surprised by this "Hessian", since the Hessian is usually never computed, the linear system can be solved without creating and storing the Hessian, but using matrix vector product.

- Could you comment on the cost in time per iteration of the proposed algorithm?
Maybe by adding the cost of each step in the algorithm?

- Once the problem is smoothed why not directly applying implicit differentiation? As in [1] or [2]?

- It seems that the algorithm introduces a lot of new hyperparameters controlling the convergence of the algorithm.
How robust/sensitive to these hyperparameters is the algorithm? In particular, the resolution can be very sensitive to the sequence $(\eta_t)$

Minor:
- It would be friendlier to indicate the number of hyperparameters tuned for each problem.


[1] Bengio, Y. (2000). Gradient-based optimization of hyperparameters. Neural computation

[2] Pedregosa, F. (2016). Hyperparameter optimization with approximate gradient. In International conference on machine learning

**Summary Of The Paper:**

The paper introduces an algorithm to solve bilevel optimization problems with non-smooth inner problems.
Authors propose to smooth the non-smooth term in the inner problem, and to iteratively decrease the smoothing parameter.
In addition, a stochastic approach is proposed to avoid compute full hypergradient.

**Summary Of The Review:**

I have concerns about how practical this algorithm is: there are so many hyperhyperparamters, I do not if it will be useful.
In addition I am not sure of how strong is the theoretical contribution.

---

### Official Review · Reviewer_KAnF · 2022-10-22

**Confidence:** 3
**Correctness:** 3
**Technical Novelty And Significance:** 2
**Empirical Novelty And Significance:** 3
**Recommendation:** 5

**Clarity, Quality, Novelty And Reproducibility:**

Clarity: The contribution looks clear and brief. This paper is generally clear and well organized. Some points need to be clarified as shown below.

(1) The meaning of the notations in eq. (1) could be provided. For example, to my understanding, $g$ is loss, $\exp(\lambda_1)$ is penalty coefficient, $\mathbf{D}_i$ is the $i$-th dataset with $r$ samples, $h$ is prediction function, $\phi$ is penalty function, right? Also, what does "$h_i$ is not Lipschitz continuous at $\mathbf{D}_i^{\top}\overline{\mathbf{w}}$" mean? Do you mean $h_i(\cdot)$ is not Lipschitz continuous in a neighborhood of $\mathbf{D}_i^{\top}\overline{\mathbf{w}}$, or $h_i(\mathbf{D}_i^{\top}\overline{\mathbf{w}})$ is not a Lipschitz continuous function of $\mathbf{w}\in\mathbb{R}^d$?

(2) In the OSCAR penalty in Section 2.2, what do $\hat{\lambda}$, $d$.
$\check{\lambda}$ mean? What is the maximum of two vectors, which seems to be a vector not a scalar we want?

(3) You could explicitly write the formula for $\widehat {\nabla} _ { \mathbf{\lambda} } \mathcal{L}$. In Algorithm 2, you could cite the numbered equations or write inline equations to show how to calculate the required variables. For example, "Calculate the stochastic gradient $\widehat{\nabla} _ {\mathbf{p}}\mathcal{L}(\ldots)$ using eq. (??)", "Calculate $A_{t,j}=\text{diag}\big(\sqrt{\text{clip}(\widehat{m}_{t,1},\rho,b)}\big)$ for $j=1,2,3$", etc.

Also cite eqs. for calculating $m_{t,1}$, $\widehat{m} _ {t,1}$, $A_{t,1}$, etc.

(4) Should the input of Algorithm 2 include $\tau$?

(5) In eq. (6) about $\widehat{\nabla} _ {\mathbf{p}}\mathcal{L}(\ldots)$, should $\lambda$ be $\tau$?

(6) After eq. (6), you could tell the names of the momentum-based variance reduction (e.g. STORM, hybrid SGD) and adaptive methods you use and cite the corresponding papers.

(7) The sentence after eq. (10) seems incorrect, since in Algorithms I only see decrease in $\mu^k$ and $\epsilon^k$ but not increase in penalty $\beta$.

(8) In Definition 2, do you mean $\mathbf{\epsilon}=[\epsilon_1,\epsilon_2,\epsilon_3]$?

(9) In Definition 4, should $H(\mathbf{w})$ be $H(\mathbf{w},\mathbf{\lambda})$?

(10) The wording of Proposition 2 could be improved. For example "then $(w',\lambda',p')$ where $p'=p^*(w,\lambda)$ is the $\epsilon$-stationary point of $\min_{w,\lambda}\max_{p\in\Delta^d}\mathcal{L}(w,\lambda,p,\mu^k)$". "then $(w',\lambda')$ is a stationary point of $H(w,\lambda)$".

(11) After Remark 2, remove "," after "Before".

(12) In Lemma 1, what does $m_{t,z}$ mean? Should $\mathbf{z}=[\mathbf{w};\mathbf{\lambda}]$ be $\mathbf{z}_t=[\mathbf{w}_t;\mathbf{\lambda}_t]$? Should $\mathcal{M}_k$ be $\mathcal{M}_t$?

(13) Is Theorem 1 about convergence rate of Algorithm 2? If yes, you might replace "we have" with "Algorithm 2 has the following convergence rate".

(14) It seems that Theorems 2 and 4 can be merged into Theorem 2. For example, "then they are the stationary points of the problem (18). Furthermore, if the lower level problem in problem (1) is strongly convex, then $(\mathbf{w}^*,\mathbf{\lambda}^*)$ is the stationary point of the original nonsmooth bilevel problem."

(15) It is better to tell the full name of SPNBO in page 2, where SPNBO is mentioned for the first time.

Quality: This is a complete piece of work. The algorithm, theorems, and experiments for supporting the claims generally look reasonable.

Novelty: The novelty looks insignificant since

(Reason I) Non-Lipschitz non-smooth bi-level optimization problems have been studied previously with non-asymptotic convergence rate and/or complexity analysis, while this work only proves asymptotic convergence of the whole Algorithm 1. For example, the experiments in [1] adds strongly-concave regularizers to the lower-level objective, and [2,3] add nonconvex, non-smooth, non-Lipschitz regularizers to the upper-level objective. You may cite these papers.

[1] Ji, Kaiyi, Junjie Yang, and Yingbin Liang. "Bilevel optimization: Convergence analysis and enhanced design." International Conference on Machine Learning. PMLR, 2021.

[2] Chen, Ziyi, Bhavya Kailkhura, and Yi Zhou. "A Fast and Convergent Proximal Algorithm for Regularized Nonconvex and Nonsmooth Bi-level Optimization." ArXiv:2203.16615 (2022).

[3] Huang, Feihu, and Heng Huang. "Enhanced bilevel optimization via bregman distance". ArXiv:2107.12301 (2021).

(Reason II) There are not many application examples that fit the problem formulation in eq. (1). Why not directly use more general continuous regularizer $\phi(w)$ instead of $\phi(h(w))$?

Reproducibility: To ensure reproducibility of the experiments, it is better to tell the values used for all the inputs (hyperparameter choices) of Algorithms 1 and 2 for each experiment.

Minor comments:

(1) In contribution 1 at the end of Section 1, "which makes our method a lower time complexity.'' looks not grammatically correct. You could use for example ``which reduces the time compelxity of our method.''

(2) Right below eq. (1), $g: \mathbb{R}^d\times\mathbb{R}^{m-1}\to\mathbb{R}$, $h_i:\mathbb{R}^r\to\mathbb{R}$.

(3) In line 4 of Algorithm 1, both $\epsilon_{k}$ and $\epsilon^{k}$ are used. Unify throughout this paper.

(4) Right above eq. (5), use bolded $\mathbf{p}$. In eq. (6), you could use bolded $\mathbf{e}_j$.

(5) Assumption 2 looks unncessary as it can be inferred from the expression of $\mathcal{L}$.

**Strength And Weaknesses:**

Pros: The authors provide the first provably convergent algorithm for bilevel optimization with non-smooth non-Lipschitz lower-level function to their knowledge. The contributions look clear.

Concerns: Lack of novelty and clarity as shown in "Clarity, Quality, Novelty And Reproducibility".

**Summary Of The Paper:**

The authors provide the first provably convergent algorithm for bilevel optimization with non-smooth non-Lipschitz lower-level function to their knowledge, via smoothing and penalty techniques. The proposed algorithm is empirically more accurate and efficient than existing state of the arts.

**Summary Of The Review:**

Since the novelty is insignificant and there are many points to clarify, I recommend borderline rejection.

---

### Official Review · Reviewer_6kJf · 2022-10-31

**Confidence:** 3
**Correctness:** 3
**Technical Novelty And Significance:** 2
**Empirical Novelty And Significance:** 2
**Recommendation:** 3

**Clarity, Quality, Novelty And Reproducibility:**

- This paper was not easy to follow, and I believe the readability can be highly improved.
- Notations are sloppy. For example, some have index $k$, but some does not even though index $k$ is necessary to reduce confusion.
- The paper is also missing some important details (as stated above).


**Strength And Weaknesses:**

**Strength**
- This can handle some structured nonsmooth regularization term in the lower-level problem, which has not been studied elsewhere.

**Weaknesses**
- The title and abstract somewhat imply that the proposed method can handle general nonsmooth lower-level problem, but it is yet quite restrictive (in terms of applications), so this might seem to be an oversell.
- The SCG assumes that (10) is satisfied for any choice of $\epsilon_k$, which does not seem to be correct. In specific, Theorem 1 shows that SCG can decrease the $||\nabla H(z_t)||$ as much as possible. Let $\hat{\epsilon}$ be its tolerance after certain number of iterations. Then, this translates to $\epsilon$-stationarity of the constrained problem (3) in Definition 2, where $\epsilon_3^2$ in Definition 2 is chosen to be $\sqrt{\frac{2d\hat{\epsilon}^2 + 2d^2\tau^2}{\beta^2}}$. Since this value cannot be decreased arbitrary small for finite $\tau$ and $\beta$, it looks possible that SCG might not meet the condition (10) for any $\epsilon_k$. So, how we choose and update $\epsilon_k$, $\beta$, $\tau$ seems important, but this does not seem to be carefully discussed in the paper.
- Algorithm 1 should have a line for updating $\beta$. Overall, the algorithm is not well written down.
- Assumption 2 is already satisfied, so this seems redundant. Technically, Assumption 3 on $\mathcal{L}$ is not satisfied.

**Minor**
- page 1: been
- page 3: OSCAR penalty not defined properly; the dimension of $w$ "is large";
- page 4: SCG samples the constraint, rather than an element $w_i$; distribution $p_{t+1}$
- page 5: $\beta^k$? How about $\tau$ as iteration goes?
- page 6: Lemma 1: $||\nabla H(z_t)||$; Theorem 1: $A_{t+1,1}$, rather than $a_{t+1,1}$? $m$ without index?

**Summary Of The Paper:**

This paper considers a specific nonsmooth bilevel problem with a structured nonsmooth regularization term, such as the $\ell_p$ norm and the OSCAR penalty, in the lower-level problem. The authors first recursively smooth the nonsmooth regularization term with a smoothing parameter $\mu^k$ that is set to decrease to $0$ as iteration goes. Each corresponding smoothed bilevel problem is solved by first transforming it into an equivalent single-level problem with a constraint that enforces the optimality of the smoothed lower-level problem. This single-level problem is further transformed into a regularized minimax problem via the penalty technique, which is then solved by the (partially stochastic) gradient (descent-ascent) method. The authors finally show that the sequence of approximate solutions (for each $\mu^k$) converges to a point satisfying a necessary optimality condition.

**Summary Of The Review:**

This paper proposed an iterative method that finds a solution of a bilevel problem with some structured nonsmooth regularization term in the lower-level problem, which has not been studied elsewhere. Although it has some merits, the paper is missing some details, such as how $\beta$ is updated in accordance to $\epsilon$, which makes it hard to verify the correctness of the analysis.

---

### Official Review · Reviewer_rqtu · 2022-11-02

**Confidence:** 3
**Correctness:** 3
**Technical Novelty And Significance:** 2
**Empirical Novelty And Significance:** 3
**Recommendation:** 5

**Clarity, Quality, Novelty And Reproducibility:**

Most parts of the paper are clear. The authors claim l_1 norm is not Lipschitz. That is confusing. I mean, l_1 norm is not smooth but how come it is non-Lipschitz. The Lipchitz constant for l_1 norm is just 1, right? In addition, the definition of Clarke subdifferential seems to require the function to be locally Lipschitz in the first place. So when the authors use the terminology "non-Lipschitz," what is the real meaning? Being not global Lipschitz but locally Lipschitz?

The algorithm looks interesting but overly complicated. Similar for the analysis. I think there is some novelty buried in the complicated mathematical derivations. It will be a significant improvement if the authors can make their proof more transparent.

**Strength And Weaknesses:**

Strength:
1. Some new analysis has been provided in this paper to provide some interesting convergence guarantees.

2. The numerical result looks promising.



Weakness:
1. The proposed algorithm looks quite overly complicated with a lot of hyperparameters.

2. The proof is also quite complicated and hard to follow. I was not able to verify the correctness of the proof.

**Summary Of The Paper:**

This paper proposes a new nonsmooth bi-level optimization algorithm based on smoothing and penalty techniques. New convergence conditions are derived for problems which may even have non-Lipschitz lower-level problem. Some numerical experiments are provided to demonstrate the effectiveness of the proposed method.

**Summary Of The Review:**

The paper seems to have some interesting algorithm/analysis developments. However, it is very hard to follow the mathematical proofs in the paper. The algorithm also seems overly complicated with many hyperparameters. For these reasons, I feel the paper is slightly below the bar.

---

### Decision · Program_Chairs · 2023-01-20

**Decision:**

Reject

**Justification For Why Not Higher Score:**

This is a  paper with rather limited novelty.
It introduces a lot a new hyperparameters controlling the convergence of the algorithm (too many hyper-hyperparamters).
The theoretical contribution seems limited though.




**Justification For Why Not Lower Score:**

The numerical result looks promising.

**Metareview: Summary, Strengths And Weaknesses:**

The paper introduces an algorithm to solve bilevel optimization problems with non-smooth inner problems.
The authors propose to smooth the non-smooth term in the inner problem, and to iteratively decrease the smoothing parameter. In addition, a stochastic approach is proposed to avoid computing the full hypergradient.